# Direct measurement of conformational strain energy in protofilaments curling outward from disassembling microtubule tips

Jonathan W Driver[1], Elisabeth A Geyer[2,3], Megan E Bailey[1], Luke M Rice[2,3]*, Charles L Asbury[1]*

[1]Department of Physiology and Biophysics, University of Washington, Seattle, United States; [2]Department of Biophysics, UT Southwestern Medical Center, Dallas, United States; [3]Department of Biochemistry, UT Southwestern Medical Center, Dallas, United States

**Abstract** Disassembling microtubules can generate movement independently of motor enzymes, especially at kinetochores where they drive chromosome motility. A popular explanation is the 'conformational wave' model, in which protofilaments pull on the kinetochore as they curl outward from a disassembling tip. But whether protofilaments can work efficiently via this spring-like mechanism has been unclear. By modifying a previous assay to use recombinant tubulin and feedback-controlled laser trapping, we directly demonstrate the spring-like elasticity of curling protofilaments. Measuring their mechanical work output suggests they carry ~25% of the energy of GTP hydrolysis as bending strain, enabling them to drive movement with efficiency similar to conventional motors. Surprisingly, a $\beta$-tubulin mutant that dramatically slows disassembly has no effect on work output, indicating an uncoupling of disassembly speed from protofilament strain. These results show the wave mechanism can make a major contribution to kinetochore motility and establish a direct approach for measuring tubulin mechano-chemistry.

*For correspondence: Luke.Rice@ UTSouthwestern.edu (LMR); casbury@uw.edu (CLA)

**Competing interests:** The authors declare that no competing interests exist.

## Introduction

Microtubules are protein polymers that grow and shorten by addition and loss of $\alpha\beta$-tubulin subunits from their tips (reviewed in *Desai and Mitchison, 1997*). In addition to supporting cell structure and serving as tracks over which motor enzymes move, the filaments can act more directly to produce force and movement – that is, to do mechanical work – independently of motor enzymes. Microtubule polymerization can generate pushing forces (*Dogterom and Yurke, 1997*; *Janson et al., 2003*) and depolymerization can generate pulling forces (*Coue et al., 1991*; *Koshland et al., 1988*; *Lombillo et al., 1995*). An important example of microtubule pulling occurs at kinetochores, where disassembling microtubule tips drive mitotic chromosome movements (*Desai and Mitchison, 1997*; *Inoué and Salmon, 1995*; *McIntosh et al., 2010*). Similar depolymerization-driven pulling might occur at other cellular locations as well, for example at the cell cortex, where disassembling tips might generate pulling forces to position the spindle in the cell, (*Laan et al., 2012*; *Nguyen-Ngoc et al., 2007*; *Carminati and Stearns, 1997*; *Kozlowski et al., 2007*) or at spindle poles, where they might drive poleward microtubule flux (*Waters et al., 1996*). The mechanical work that a disassembling microtubule tip exerts on an isolated kinetochore, or on a collection of kinetochore sub-complexes, can be directly measured in vitro (*Volkov et al., 2013*; *Akiyoshi et al., 2010*). But the mechanism underlying this force production is unknown.

**eLife digest** Before a cell divides it must separate its chromosomes, the ribbons of DNA that carry its genes. To do this, filaments called microtubules attach by their ends to the chromosomes and then shorten, pulling the chromosomes to opposite sides of the cell. The microtubules are made of thousands of subunits packed together to form miniature tubes, and shorten by losing subunits from their ends.

Why don't the chromosomes simply fall off the ends of these microtubules, which are crumbling under their grip? How can a crumbling filament exert a pulling force? The shape of the ends of the microtubules suggests a possible answer. The subunits that make up each microtubule are arranged in rows, called protofilaments, that run along the length of the microtubule. When a microtubule shortens, its protofilaments first curl outward from the end and then crumble apart. If the curling protofilaments are strong enough, they could act like springs, hooking the chromosome and pulling on it as they curl outward.

Curling protofilaments can exert some pulling force, but how much force was not known. To investigate, Driver et al. used an instrument called a laser trap, or laser tweezers, to record tiny movements and forces exerted by individual microtubules on microscopic plastic beads. The microtubules came from yeast cells, and had been engineered to carry a tag on their surface that enabled them to attach to the beads in a way that did not interfere with the curling action of the protofilaments. The experiments revealed that curling protofilaments do indeed behave like strong springs, and can make a major contribution to moving chromosomes.

Fully understanding how microtubules pull on chromosomes could help to design anti-cancer drugs that prevent cells from dividing. Drugs that target microtubules are already used against certain cancers, but they cause considerable side effects because microtubules are important in many types of cells. However, drugs that specifically prevent curling protofilaments from tugging on chromosomes could potentially treat cancer with fewer side effects. It remains to be seen whether such drugs can be developed.

Two classes of models have been proposed to explain how disassembling microtubules produce force, conformational wave and biased diffusion (*Koshland et al., 1988*; *Hill, 1985*; *Asbury et al., 2011*). The central tenet of the conformational wave model is that individual rows of tubulin subunits, the protofilaments, pull on the kinetochore as they curl outward from a disassembling microtubule tip. Strain energy is trapped after GTP hydrolysis in the microtubule lattice, because intrinsically curved GDP-tubulin subunits are held in a straight (i.e., strained) configuration by their binding to neighboring subunits (*Desai and Mitchison, 1997*; *Caplow and Shanks, 1996*). This stored strain energy is released during tip disassembly, when the protofilaments curl outward from the tip and break apart, forming a conformational wave that propagates down the microtubule (*Mandelkow et al., 1991*; *Nogales and Wang, 2006*). Kinetochores are proposed to harness this wave to produce useful mechanical work. The central tenet of the alternative biased diffusion model is that the energy of interactions between a kinetochore and a microtubule creates a thermodynamic force that pulls the kinetochore toward the microtubule tip, analogous to the interfacial forces that draw fluid into a capillary (*Hill, 1985*; *Asbury et al., 2011*). These two models are not mutually exclusive and, in principle, a purely biased diffusion-based mechanism could operate independently of any spring-like action of the protofilaments.

The conformational wave mechanism, however, requires curling protofilaments to generate powerful 'working strokes'. A seminal study by Grishchuk and co-workers used a laser trap assay to show that disassembling microtubule tips can exert brief pulses of force against an attached bead (*Grishchuk et al., 2005*). Their analysis suggested that the conformational wave might be capable of generating very high forces, up to ~50 pN, but the actual measured forces were much lower (<0.5 pN) and probably were restricted by interference of the attached beads with the short working strokes of the protofilaments. Displacement amplitudes were not reported. Because of these limitations, the energy carried by the curling protofilaments was not determined.

Fundamentally, the work output of the conformational wave mechanism must be limited by the amount of curvature strain energy carried by GDP-protofilaments, which dictates how forcefully they can curl outward from the tip. Convincing measurements of protofilament strain energy should therefore reveal how efficiently they can produce mechanical work via the wave mechanism. Moreover, protofilament strain is fundamental to all current models of microtubule dynamic instability, and it is generally thought to drive rapid disassembly (*Desai and Mitchison, 1997*; *Nogales and Wang, 2006*; *VanBuren et al., 2005*; *Molodtsov et al., 2005*). Thus, measuring the strain energy in curling protofilaments will also provide insight into the basic mechano-chemistry of tubulin.

Based on the pioneering work of *Grishchuk et al. (2005)* we have developed a modified 'wave assay' that overcomes limitations inherent to their study. Interference from the attached bead was minimized by using recombinant tubulin with an engineered, flexible tether. By applying a feedback-controlled laser trap, nm-scale displacements were measured as functions of force, enabling direct observation of the spring-like elasticity of curling protofilaments and showing that they carry a substantial fraction of the energy of GTP hydrolysis in the form of curvature strain. To probe the relationship between strain energy and disassembly rate, we measured the wave energy of a slow-disassembling tubulin mutant. Surprisingly, a 7-fold decrease in disassembly rate had no effect on conformational wave energy, which reveals that the speed of disassembly can be uncoupled from curvature-derived protofilament strain. We present a simple model to explain how strain energy and disassembly speed can be uncoupled.

## Results

### Modified assay improves detection of conformational wave-driven movement

The prior laser trap study demonstrated for the first time that disassembling microtubule tips can exert brief pulses of force on microbeads attached to the filaments by strong inert linkers, such as biotin-avidin (*Grishchuk et al., 2005*). However, pulses were detected in fewer than half of the trials, pulse durations varied over 300-fold, and relaxation of the beads into the center of the trap was slower after the trials that failed to produce pulses. These observations suggest that the bead-microtubule attachments, which consisted of multiple biotin-avidin bonds (approximately 3 to 8), restricted outward curling of the protofilaments. Moreover, because a fixed trap was used without feedback control, pulse amplitudes were probably limited by the maximum distance over which the curling protofilaments could exert force (i.e., by their working stroke length), rather than by their total capacity for work output. These limitations made it difficult to quantitatively assess the force generating potential of the system. We therefore sought to improve the assay by developing a single molecule tethering scheme and by using a feedback-controlled trap.

To begin our modified wave assay, we grew dynamic microtubule extensions from coverslip-anchored seeds. The extensions were assembled from recombinant yeast $\alpha\beta$-tubulin, with a His$_6$ tag engineered onto the C-terminal tail of the $\beta$ subunit. Microbeads were tethered to the sides of individual, growing filaments via single anti-His antibodies, creating a strong yet flexible tether (~36 nm in length; see Materials and methods). A bead-microtubule assembly was held in the laser trap (*Figure 1a and b*) and feedback control was initiated to apply a constant tension, which reduced Brownian motion and facilitated detection of microtubule-driven movements. The distal microtubule plus end was then severed with laser scissors to induce disassembly (*Franck et al., 2010*). When the disassembling tip reached the bead, it generated a brief pulse, during which the bead first moved against the force of the laser trap, then relaxed back toward the trap center, and finally detached as the microtubule disassembled past the tether (*Figure 1c-e*). At low opposing force, a pulse was nearly always observed (90%, or 148 of 164 events recorded at <5 pN). The pulses were large, often >60 nm (*Figure 1d and e*), which is more than twice the width of the microtubules. These observations show that disassembling tips can generate pulses of movement more reliably than previously observed. The pulses were also fast, with average risetimes between 0.1 and 0.3 s (depending on the level of force; *Figures 1d,e* and *2a–b*), which is 5- to 10-fold faster than in the previous recordings. These observations suggest that our modified tethering scheme imposed less restriction on the outward curling of the protofilaments.

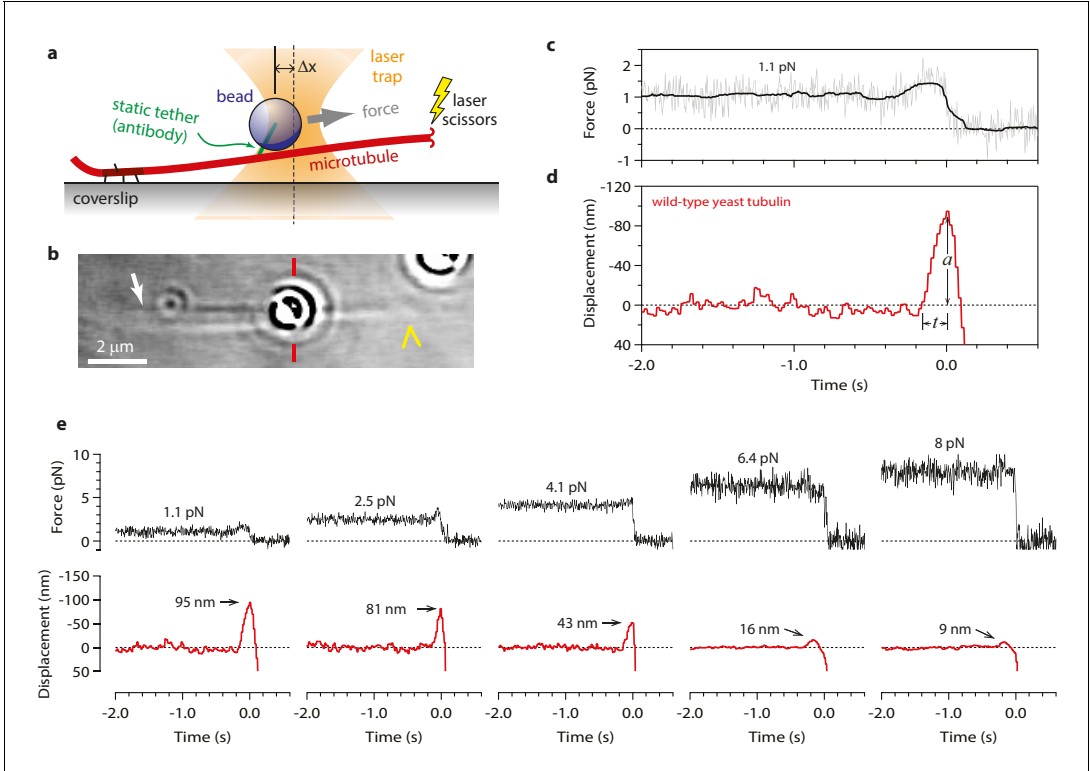

**Figure 1.** Measuring the tubulin conformational wave with a feedback-controlled laser trap. (a) A bead is tethered to the side of a microtubule via a single antibody bound to the C-terminal tail of β-tubulin and placed under tension using the laser trap. The trap is feedback-controlled to keep a fixed separation from the bead (Δx), thereby maintaining a constant level of tension. Microtubule disassembly is induced by cutting the tip with a second laser. (b) Video-enhanced differential interference contrast (VE-DIC) image of a 900 nm bead tethered to a single microtubule under laser trap tension (from *Video 1*). Approximate locations for the coverslip-anchored portion of the microtubule (white arrow), the laser trap center (red dashes), and the plus end tip (yellow chevron) are indicated. (c, d) Example record showing trap force (c) and bead displacement (d) versus time. Grey trace shows raw bead-trap separation after converting to force by multiplying by the trap stiffness. Black trace shows same data after smoothing with a 250 ms median filter. When the disassembling tip arrives at the bead, the bead initially moves against the trapping force and then releases as the microtubule disassembles out from underneath it. The pulse amplitude, *a*, and risetime, *t*, are indicated. (e) Gallery of additional example records, measured at the indicated levels of tension. Data in (c - e) were collected using 900 nm beads.

## Conformational waves can drive movement against large opposing loads

Our modified wave assay enabled us to measure pulse properties as functions of force for the first time. Pulse amplitudes decreased as the force of the laser trap was increased (*Figure 2c and d*). This behavior demonstrates directly that curling protofilaments exhibit spring-like elasticity. Eventually a 'stall force' was reached, at which the pulses were completely suppressed (*Figure 2c*). Depending on bead size, the stall force ranged from 8 to 16 pN (*Figure 2—figure supplement 1*), which is at least 16-fold higher than the maximal force measured in the previous study (<0.5 pN). The increased force production may be explained by our use of a force clamp, by our less restrictive tethering scheme, or by a combination of these two factors. It is also formally possible that the force generating capacity of microtubules grown from yeast tubulin (used here) is intrinsically higher than that of microtubules grown from bovine brain tubulin (used in the previous study). However, we consider this possibility unlikely because the shapes and lengths of curling protofilaments are very similar in yeast and vertebrate cells (*McIntosh et al., 2013*) and because, at the level of tubulin structure, the internal curvature of unpolymerized αβ-tubulin (i.e., the rotation required to superimpose α- onto β-tubulin) is also very similar (*Ayaz et al., 2014*, *2012*). In any case, our results show that protofilaments curling outward from a disassembling microtubule tip behave like springs and can generate forces much higher than previously recorded.

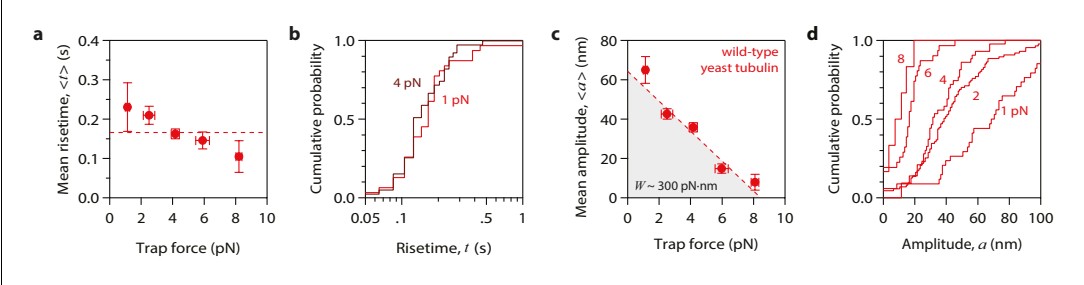

**Figure 2.** Tubulin waves generate large forces. (**a, b**) Mean pulse risetime versus force (**a**) and distributions of risetime at indicated forces (**b**) for wild-type microtubules. The mean risetime across all forces is depicted by the dashed line in (**a**). (**c, d**) Mean pulse amplitude versus force (**c**) and distributions of amplitude at indicated forces (**d**) for pulses generated by wild-type yeast microtubules. Total pulse energy, $W$, is estimated from the area under the line-fit in (**c**), shaded grey. Error bars show standard errors (for $N = 6$ to 87 amplitudes; $N = 3$ to 78 risetimes). All data in (a - d) were collected using 900 nm beads.

The following figure supplement is available for figure 2:

**Figure supplement 1.** Properties of wild-type tubulin waves measured using different bead sizes.

## Mechanism of wave-driven movement: protofilaments push laterally, bead pivots about tether

Measurements of wave-driven bead movement can potentially be used to estimate the total capacity of the conformational wave for mechanical work output, provided the mechanism underlying movement in the assay is understood. Beads in our assay were linked to the microtubules through the flexible C-terminal tails of $\beta$-tubulin. Flexible tethering implies that when a microtubule-attached bead is placed under tension, the tether should become extended and the bead surface should initially rest against the microtubule wall at a secondary contact point (**Figure 3a**). Starting from this initial condition, we considered two scenarios for how the pulses of bead movement might be generated. In the 'lateral push' scenario, the curling protofilaments push laterally against the bead at the secondary contact point, causing the bead to pivot about the base of the tether (**Figure 3b**). The bead acts as a lever in this case, but because the fulcrum is located at the tether, away from where the curling protofilaments exert their force, the predicted leverage is only modest (~2 fold, depending on bead size and tether length). In the second scenario, 'axial pull', the microtubule first disassembles past the secondary contact point,

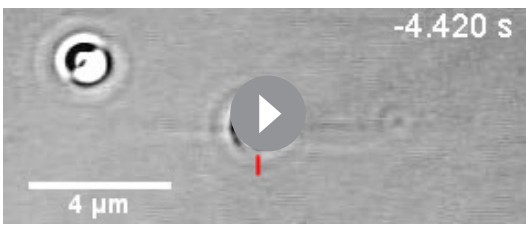

**Video 1.** Example of wave assay. A bead tethered to the side of a coverslip-anchored microtubule is initially held under laser trap tension (here, ~1 pN). The distal plus end of the microtubule is severed by laser scissors (at 0 s), triggering disassembly. When the disassembling end reaches the bead, it causes a brief pulse of motion (0.7 s) before the bead detaches (1.0 s). After bead detachment, the microtubule continues disassembling while the stage also moves rightward under feedback control. Red dashes mark the approximate location of the center of the laser trap.

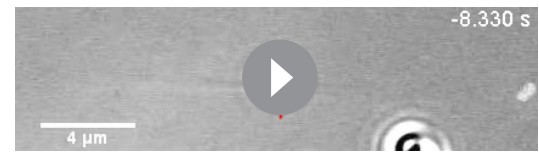

**Video 2.** Second example of wave assay. A bead tethered to the side of a coverslip-anchored microtubule is initially held in the laser trap, at low tension (<1 pN). Feedback control is initiated (at −5.6 s) to apply higher tension (4 pN), and then the distal plus end of the microtubule is severed (0 s). The bead detaches when it is reached by the disassembling end (1 s). After bead detachment, the microtubule continues disassembling while the stage moves rightward under feedback control. Red dashes mark the approximate location of the center of the laser trap.

allowing the bead to rotate under laser trap tension into an end-on configuration relative to the microtubule tip (*Figure 3c*). Then the working stroke occurs when curling protofilaments encounter the tether and pull axially on the bead (*Figure 3d*). There is no leverage in this case. The unamplified trapping force opposes protofilament curling directly.

Because of the relatively large bead radius, its rotation into an end-on configuration would produce an obvious relaxation toward the trap center, which in the axial pull scenario must precede the working stroke by ~200 ms (the time required for tip disassembly to propagate from the secondary contact point to the tether). However, we found that the bead position was nearly always stable prior to the pulses (*Figure 1e*), except in a very small fraction of trials (~2%, 18 of 760) during which the initial pulse, from a stable baseline, was followed by bead relaxation toward the trap center and then by a second pulse (*Figure 3—figure supplement 1*). These rare secondary pulses might be generated by axial pulling. However, the lack of any relaxation before the primary pulses indicates that these were not preceded by rotation into an end-on configuration, and thus were not generated by axial pulling. Thus, it seems that the lateral push mechanism underlies bead movement in most cases.

To test more directly whether the lateral push model was operational, we examined how pulse amplitudes varied with laser trap tension and bead size. Altering bead size is predicted to have two consequences. First, larger beads should increase leverage and therefore decrease the amount of laser trap tension required to suppress the pulses. Consistent with this prediction, stall forces decreased from 16.2 ± 3.0 pN to 8.4 ± 0.9 pN as bead diameter was increased from 320 to 900 nm (*Figure 4a*). The second prediction, also a consequence of leverage, is that larger beads should produce larger pulse amplitudes when the opposing tension is low enough to allow unhindered movement. Indeed, the maximum pulse amplitudes, extrapolated to zero tension, increased from 45.2 ± 3.6 nm to 64.3 ± 3.6 nm as bead diameter was increased from 320 to 900 nm (*Figure 4b*). The relationships for stall force-vs-bead diameter and for unloaded amplitude-vs-bead diameter can be predicted quantitatively from simple geometric considerations, given estimates of the height that the curls project from the microtubule surface (~20 nm, based on electron micrographs of disassembling tips) (*Mandelkow et al., 1991*; *McIntosh et al., 2013*, *2008*) and of the tether length (~36 nm; see Materials and methods). The predicted curves fit our data well, and they are relatively insensitive to the precise tether length (*Figure 4a and b*), suggesting that the lateral push model provides a good description of the underlying mechanism.

## Conformational waves carry substantial amounts of strain energy

Measuring pulse amplitude as a function of trapping force enabled us to calculate the total mechanical work output of the assay, $W$, which is given by the area under the amplitude-vs-force curve (e.g., *Figure 2c* and *Figure 2—figure supplement 1*). Whereas changes in bead size altered the amplitude-vs-force curve in predictable ways (as discussed above), the total work output was independent of bead size (*Figure 4c*). This invariance would not be expected if work output was limited by the attached bead, and therefore it suggests that $W$ indeed measures the *intrinsic* strain energy carried by the curling protofilaments that pushed against the bead. Based on a global average across all three bead sizes, we estimate $W = 304 ± 24$ pN·nm (*Figure 4c*), a value 74-fold greater than thermal energy ($k_BT$). Assuming the lateral push model correctly describes the assay geometry, a maximum of 4 curls could push simultaneously against the beads (*Figure 4—figure supplement 1b*). Given the 23° curvature and 8 nm length of an individual tubulin dimer, (*Mandelkow et al., 1991*; *Amos and Klug, 1974*) the estimated curl height of $h = 20$ nm further suggests that the curled segments are ~4 dimers in length (*Figure 4—figure supplement 1a*). Thus, the total work output, $W$, may derive from outward curling of as many as 16 tubulin dimers, implying that the wave carries at least 19 pN·nm of energy per dimer (4.7 $k_BT$, or 2.7 kcal mole$^{-1}$; *Figure 4—figure supplement 1c*). These observations establish that the conformational wave carries considerable strain energy that can be harnessed to perform mechanical work, and they provide a direct estimate of the strain per tubulin subunit.

## Disassembly speed can be uncoupled from curvature strain

Mechanical strain in the microtubule lattice is commonly assumed to drive the rapid disassembly of microtubules, (*Desai and Mitchison, 1997*; *Nogales and Wang, 2006*; *VanBuren et al., 2005*;

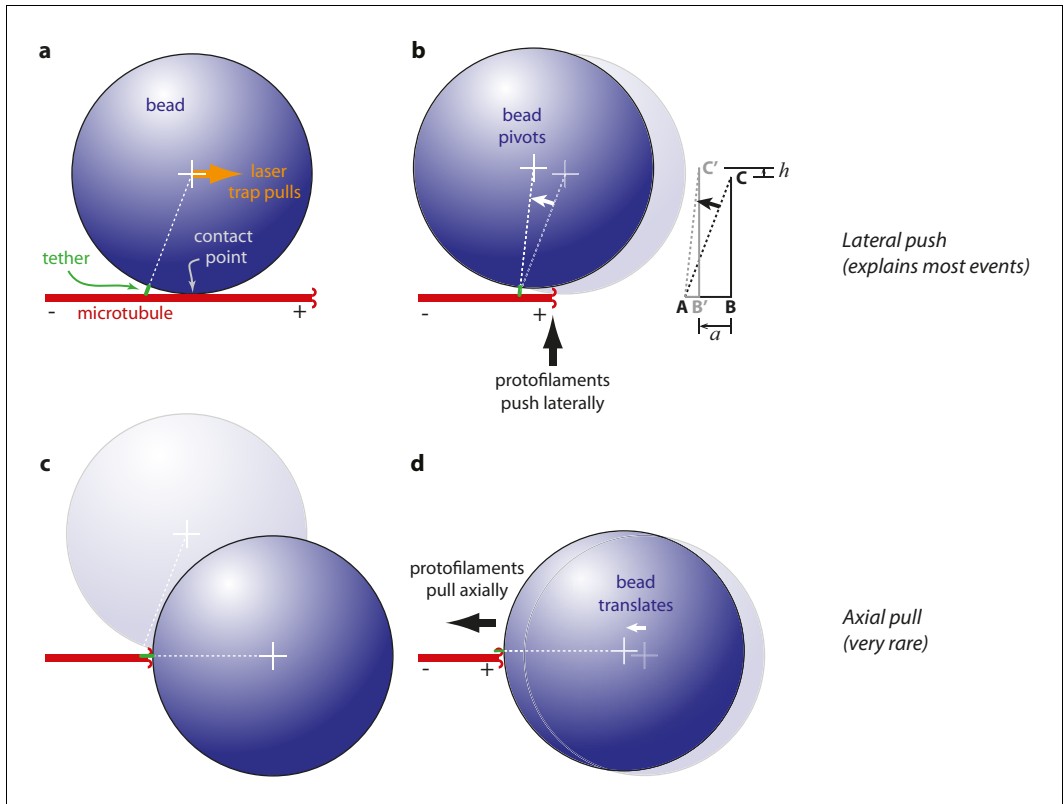

**Figure 3.** Proposed mechanisms underlying conformational wave-driven bead movement in the assay. (**a**) Initially, when a bead is placed under tension it rests against the microtubule wall at a secondary contact point. (**b**) In the lateral push scenario, the curling protofilaments push laterally against the secondary contact point, causing the bead to pivot about the base of the tether. Lateral deflections from the protofilaments, *h*, produce larger axial displacements of the bead, *a*. If A is the tether point and B is the point of bead-microtubule contact and C is the bead center, then ABC defines a right triangle and the amount of leverage is given by the ratio of sides BC/AB. The predicted leverage for 900 nm diameter beads attached via 36 nm tethers is $a \cdot h^{-1} = 2.4$. (**c**) In the axial pull scenario, the microtubule first disassembles past the secondary contact point, allowing the bead to rotate under laser trap tension into an end-on configuration relative to the microtubule tip. (**d**) Then the working stroke occurs when curling protofilaments encounter the tether and pull axially on the bead.

The following figure supplement is available for figure 3:

**Figure supplement 1.** A rare example record in which the initial pulse, from a stable baseline, was followed by bead relaxation toward the trap center and then by a second pulse (double arrow).

*Molodtsov et al., 2005*) but it has not been possible to test directly for a relationship between lattice strain and disassembly speed. To determine whether curvature strain dictates the speed of microtubule disassembly, we used our assay to measure the wave energy for microtubules assembled from a slow-shortening mutant tubulin. A number of tubulin mutations that 'hyperstabilize' microtubules in vivo have been described, (*Geyer et al., 2015*; *Gupta et al., 2002*; *Machin et al., 1995*) and some have been shown to slow the disassembly of microtubules grown in vitro from purified mutant tubulin (*Geyer et al., 2015*; *Gupta et al., 2002*; *Sage et al., 1995*). We focused on a particular mutant in which threonine 238 of β-tubulin is replaced by valine (T238V; see *Figure 5a*) (*Geyer et al., 2015*). The mutated residue is buried inside β-tubulin, where it cannot directly perturb inter-subunit lattice contacts. T238V tubulin forms microtubules that disassemble 7-fold more slowly than wild-type (*Figure 5—figure supplement 1*). If the rate of disassembly is determined by lattice strain, then the slow-disassembling mutant microtubules should store less conformational strain energy than their wild-type counterparts.

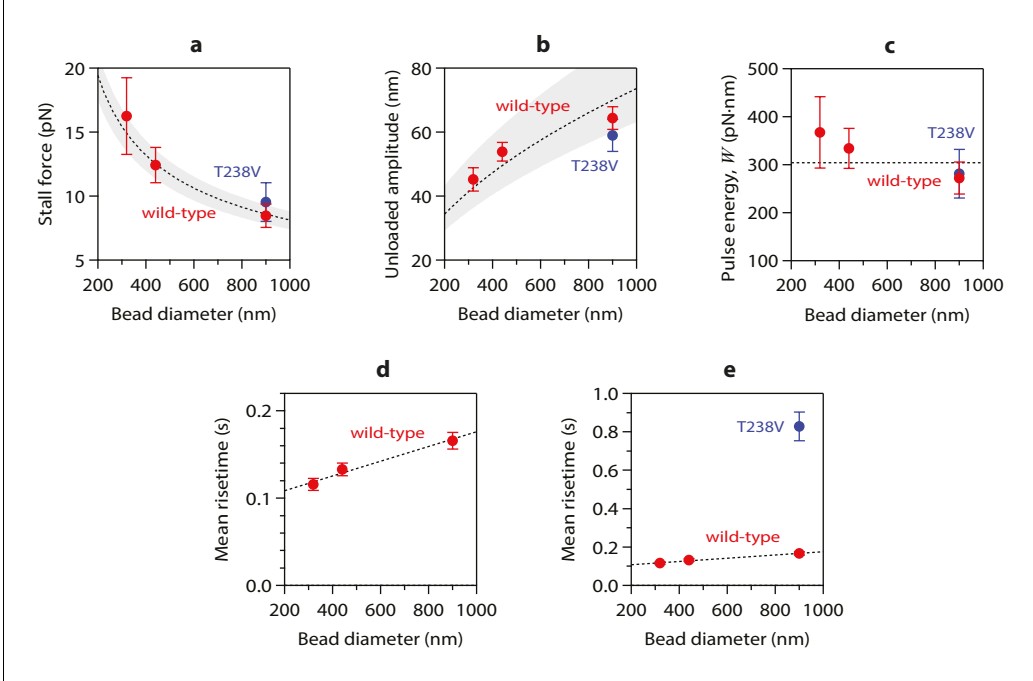

**Figure 4.** Stall forces and pulse amplitudes vary with bead size, but pulse energy is invariant. (**a**) With increasing bead size, the leverage increases and therefore the trapping force required to completely suppress the pulses (i.e., the 'stall force') decreases. (**b**) Unloaded pulse amplitudes (i.e., amplitudes extrapolated to zero tension) increase with bead size, because the amplification ratio increases (see *Figure 3b*). Dotted curves in (**a**) and (**b**) show predictions assuming a tether length of 36 nm and a curl height, $h$ = 20 nm. Gray shaded regions show predicted ranges for tether lengths ranging from 30 to 42 nm. (**c**) The total pulse energy, $W$, is independent of bead size. Horizontal dotted line in (**c**) shows global estimate of pulse energy, $W$ = 304 ± 24 pN·nm, from a weighted fit of the wild-type data across all bead sizes. (**d, e**) Mean pulse risetimes as a function of bead size. Wild-type data in (**d**) are replotted in (**e**) with an expanded scale for comparison to the mutant, T238V.

The following figure supplement is available for figure 4:

**Figure supplement 1.** Estimation of strain energy per tubulin.

Contrary to this prediction, however, the conformational wave energy for T238V microtubules was indistinguishable from that of wild-type microtubules. T238V microtubules in the wave assay produced pulses that were ~5 fold slower than wild-type (*Figures 4e*, *5c,d* and *6a*), consistent with their slower disassembly speed. However, the wave amplitude-vs-force relation for T238V microtubules was essentially identical to that of wild-type (*Figure 6c*). Thus, even though they disassemble at very different rates, wild-type and T238V microtubules must store similar amounts of curvature-derived mechanical strain (*Figure 4c*). The similar amplitude-vs-force curves further suggest that the mutation does not substantially alter the intrinsic curvature, flexural rigidity, or contour length of protofilament curls, because all of these properties together determine pulse amplitude.

## Energy landscape model for the curling reaction

How might the rate of microtubule disassembly be uncoupled from mechanical strain in the lattice? To begin addressing this question, we developed a simple model for the energy landscape of a single GDP-tubulin dimer curling outward from a disassembling tip (*Figure 7*). The mechanical strain energy carried by the dimer was modeled as a function of its bend angle by assuming it behaves as a slender elastic rod with a naturally bent shape, with 23° of curvature when fully relaxed, (*Mandelkow et al., 1991*) and with ~5 $k_BT$ of strain at 0° curvature. (Its flexural rigidity was chosen such that the fully straightened dimer carries a strain energy similar to our estimated value, 19 pN·nm; see *Figure 7—figure supplement 1a*.) The energy of the lateral bonds the dimer forms with

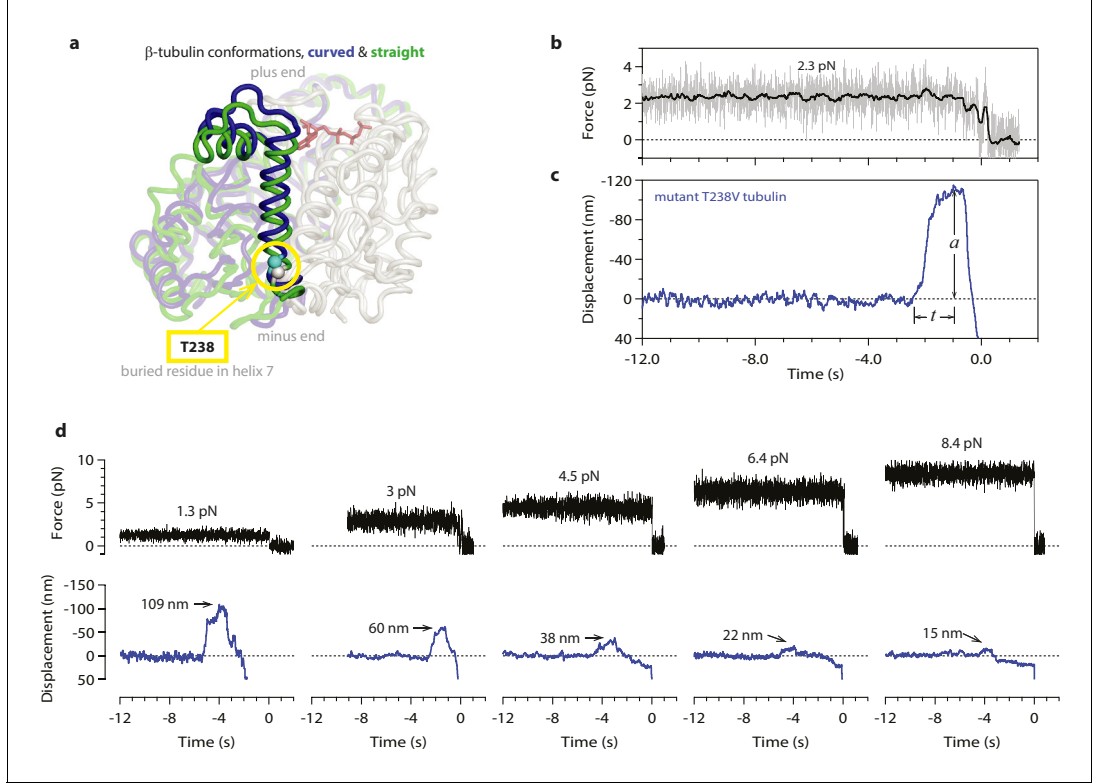

**Figure 5.** Hyperstable mutant microtubules produce slower pulses. (a) Superposition of polymerized ('straight', green) and unpolymerized ('curved', blue) conformations of β-tubulin. Residue T238 is inaccessible to solvent and located on a helix (H7) that undergoes piston-like movement between the straight and curved conformations (which are represented by PDB entries 3JAT and 1SA0, respectively). GDP nucleotide is shown in red. (b, c) Example record showing trap force (b) and bead displacement (c) versus time for a mutant T238V microtubule. Grey trace shows raw bead-trap separation after converting to force by multiplying by the trap stiffness. Black trace shows same data after smoothing with a 250 ms median filter. The pulse amplitude, *a*, and risetime, *t*, are indicated. (d) Gallery of additional example records for mutant T238V microtubules, measured at the indicated levels of tension. Data in (b - d) were collected using 900 nm beads. Note the different time scales here in comparison to *Figure 1c–e*.

The following figure supplement is available for figure 5:

**Figure supplement 1.** Hyperstable mutant T238V tubulin disassembles more slowly than wild-type.

its neighbors in the microtubule wall was assumed to follow a simple (Lennard-Jones) function of the bend angle (*Figure 7—figure supplement 1b*). These mechanical strain and lateral bond energies were added together to calculate a total free energy landscape (*Figure 7—figure supplement 1c*). The predicted landscape implies a curling reaction that proceeds via a high-energy transition state. We envision that the lateral bonds are short-range interactions, such that they break before much curling has developed. With this assumption, the high-energy transition state should closely resemble the initial, straight conformation (and the curling reaction can be considered 'Eyring-like' [*Howard, 2001*]).

According to this model, the slower disassembly of T238V microtubules is explained by an increase in the height of the activation barrier, which could arise either because the energy of the transition state is higher or because the energy of the starting state (i.e., when the tubulin is straight and laterally bonded) is lower. Our data exclude the possibility of a substantially higher transition state energy because this would lead to a higher wave energy for the mutant, which we did not observe. We therefore propose that the mutation specifically strengthens lateral bonds, thereby lowering the energy of the starting state and raising the activation barrier, without altering the intrinsic 23° curvature or the mechanical rigidity of the dimer (*Figure 7b*).

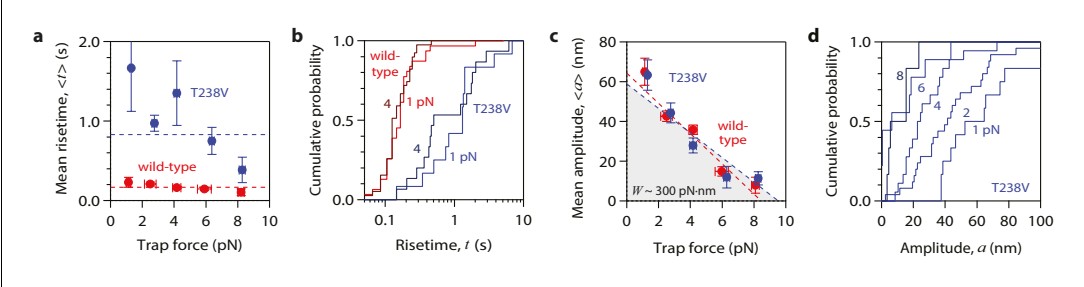

**Figure 6.** Hyperstable mutant microtubules produce pulses with identical energy. (a) Mean pulse risetime versus force for mutant T238V microtubules. Wild-type data (from *Figure 2a*) is shown for comparison. The mean risetimes across all forces for T238V and wild-type microtubules are depicted by the dashed blue and red lines, respectively. Error bars show standard errors (for $N = 6$ to 25 amplitudes; $N = 2$ to 78 risetimes). (b) Distributions of risetime at indicated forces for wild-type and T238V microtubules. (c, d) Mean amplitude versus force (c) and distributions of amplitude at indicated forces (d) for pulses generated by mutant T238V microtubules. Wild-type data (from *Figure 2c*) is shown in (c) for comparison. Total pulse energy, $W = 280 \pm 50$ pN·nm, estimated from the grey-shaded area under the line-fit, is similar for both types of microtubules. All data in (a - d) were measured with 900 nm beads.

## The T238V mutation affects tubulin lattice structure and strengthens tubulin-tubulin bonds

To further explore whether the T238V mutation might strengthen lateral bonds in the microtubule lattice, we took two additional approaches. In one approach, we probed the conformation of tubulin in the lattice using the plus-end-tracking EB1-family protein, Bim1. Like other EB1 proteins, (*Zanic et al., 2009*; *Bieling et al., 2007*) Bim1-GFP brightly decorates the growing plus-ends of wild-type yeast microtubules, with a strong preference for the growing ends over the remainder of the filament lattice (*Geyer et al., 2015*). We observed similar bright Bim1-GFP decoration at the growing plus-ends of mutant T238V microtubules as well, but the lattice of the mutant T238V microtubules retained an abnormally high affinity for Bim1-GFP (*Figure 5—figure supplement 1c and d*). This observation indicates that the lattice conformation of T238V tubulin retains structural characteristics that are normally found only near the ends of growing microtubules (GTP-cap-like), which may be associated with stronger lateral bonding in the lattice.

As a second approach for examining the effects of the T238V mutation, we devised a new 'plucking' assay to measure the forces required to remove tubulins from growing microtubule ends. We fortuitously found, using the same flexible tethers devised for the wave assay (i.e., single anti-His antibodies bound to a His₆ tag on the C-terminal tail of $\beta$-tubulin), that individual beads could be linked to the growing ends (rather than the sides) of single, dynamic microtubules. If increasing tension was then applied ($0.25$ pN·s$^{-1}$), the end-bound bead could be detached (*Figure 8a and b*). End-bound beads were readily detached in this manner, but side-bound beads generally did not detach, even at the maximum laser trap tension (~40 pN under the conditions used here). Usually, detaching an end-bound bead by force triggered immediate disassembly of the microtubule (43 of 57 detachments, ~75%), which confirms that tubulin dimers were forcibly removed (*Figure 8c*). Given the single antibody-based linkages, the number of plucked dimers was probably low, but possibly greater than one or two. The average force required to pluck tubulins from a wild-type microtubule end was $8.3 \pm 0.6$ pN (*Figure 8d and f*). Plucking tubulins from mutant T238V microtubules required considerably more force, $19.0 \pm 1.6$ pN on average (*Figure 8e and f*). This higher plucking force is consistent with stronger lateral bonds, although it could arise from a strengthening of longitudinal bonds, or a strengthening of both kinds of bonds. Whether it would also occur in the context of a disassembling end remains uncertain; nevertheless, the observation shows that mutant T238V tubulin forms relatively stronger tubulin-tubulin bonds compared to wild-type, at least in the context of an assembling end.

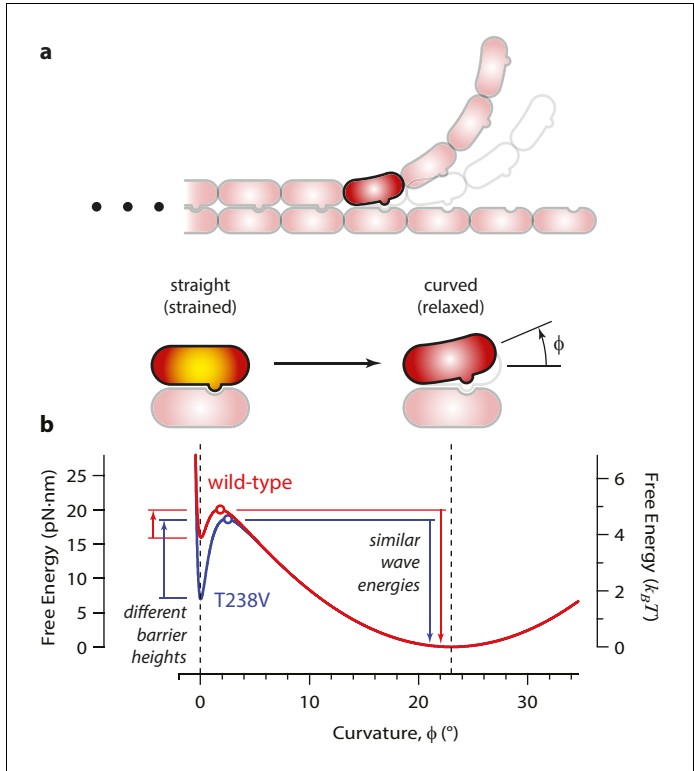

**Figure 7.** Free energy landscape for a curling αβ-tubulin. (**a**) The model considers a single αβ-tubulin (highlighted) as it bends outward from a microtubule. For simplicity, only two protofilaments are depicted. The curling subunit is shown (arbitrarily) at the base of a previously formed protofilament curl. (**b**) Hypothetical free energy landscapes for wild-type (red curve) and mutant T238V tubulin (blue curve) as functions of subunit curvature, φ. Lateral bonding initially holds the tubulin in a straight conformation (strained, φ = 0°). Curling then proceeds via a high-energy transition state (open circles), which is reached without the development of much curvature (φ ~ 2°). Stronger lateral bonding in T238V increases the height of the transition energy barrier, reducing the rate of curling relative to wild-type. Relaxation from the highly strained transition state to the naturally curved ground state (at φ = 23°, with free energy arbitrarily set to zero) drives movement in the wave assay. Because T238V and wild-type have similar transition energies, they produce conformational waves with similar energy.

The following figure supplement is available for figure 7:

**Figure supplement 1.** Free energy landscape for a single curling αβ-tubulin subunit, calculated by adding independent contributions from mechanical strain and lateral bonding.

## Discussion

As with any cantilevered spring, the amount of *force* that curling protofilaments can produce depends on how they are coupled to the object on which they are pushing (*Molodtsov et al., 2005*; *Efremov et al., 2007*). The amount of *strain energy* they carry is a more fundamental quantity, and therefore less sensitive to geometric details of the coupling. Ultimately, this strain energy determines the maximum force-generating capacity of the conformational wave mechanism. It is also fundamentally important for all current models of microtubule dynamic instability, and numerous previous studies have attempted to estimate its magnitude. Thermodynamic approaches (*Desai and Mitchison, 1997*; *Caplow and Shanks, 1996*; *Howard, 2001*) and analyses based on the bending rigidity of intact microtubules (*Mickey and Howard, 1995*) have yielded estimates spanning more than an order of magnitude (*Figure 4—figure supplement 1d*). But these methods can only infer the stored strain indirectly. Our wave assay has provided a more direct approach.

To measure the energy carried by the conformational wave we modified an assay pioneered in a previous study, (*Grishchuk et al., 2005*) adding a feedback-controlled laser trap and other

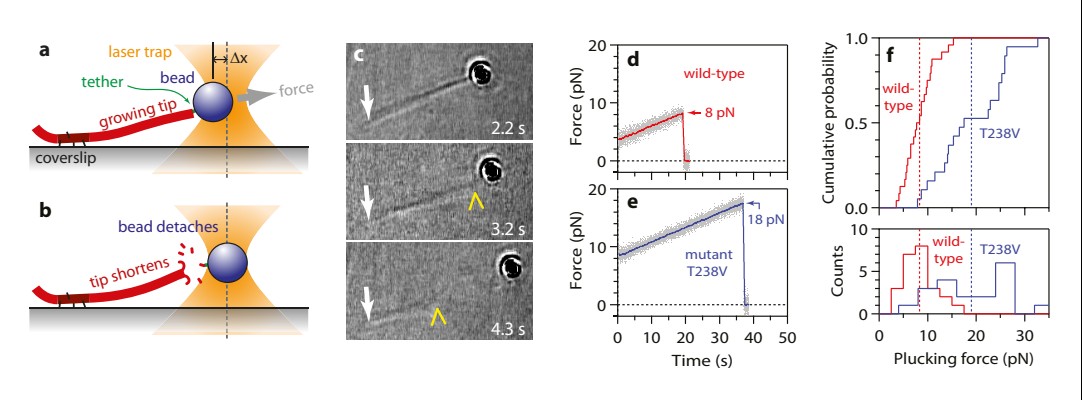

**Figure 8.** More force is required to pluck hyperstable mutant tubulin subunits from the microtubule end. (**a**) A bead is tethered to the end of a growing microtubule via a single antibody bound to the $\beta$-tubulin C-terminus and then tested with a 0.25 pN·s$^{-1}$ force ramp. (**b**) Usually, detaching the bead by force triggers immediate disassembly of the microtubule (43 of 57 detachments, ~75%), indicating that tubulin dimers were forcibly removed. (**c**) Selected frames from *Video 3*, showing an end-tethered bead under tension (2.2 s), detachment of the bead (3.2 s), and tip shortening (4.3 s). White arrows mark the coverslip-anchored segment of the microtubule. Yellow chevrons mark the plus end. (**d, e**) Example records of tensile force versus time for beads tethered to the ends of wild-type and hyperstable mutant T238V microtubules. Arrows mark plucking forces. Gray dots show raw data. Colored traces show same data after smoothing with a 500 ms boxcar average. (**f**) Distributions of plucking force for wild-type and mutant T238V tubulins. Dotted vertical lines indicate averages for wild-type tubulin, 8.3 ± 0.6 pN (mean ± SEM; $N$ = 24), and for T238V, 19.0 ± 1.6 pN ($N$ = 19).

improvements to prevent the microbeads used in the assay from restricting the curling of the protofilaments. Our results clearly demonstrate the spring-like elasticity of the protofilaments and establish that they carry a very substantial amount of curvature strain (>74 $k_BT$), which can be harnessed to perform mechanical work. Our data further show that movement in the assay is likely driven by a lateral push mechanism, in which the curling protofilaments push laterally on the bead, causing it to pivot around the flexible tether linking it to the microtubule. Based on this arrangement, we estimate that the measured work output derives from outward curling of as many as 16 tubulin dimers, implying an energy per dimer of at least 19 pN·nm (4.7 $k_BT$, or 2.7 kcal mole$^{-1}$).

Ultimately this energy is derived from the chemo-mechanical cycle of tubulin. Soon after a tubulin dimer assembles into a microtubule, GTP is hydrolyzed and a portion of the free energy liberated by this chemical reaction is stored as curvature strain in the microtubule lattice (*Desai and Mitchison, 1997*; *Nogales and Wang, 2006*). Our estimate of this stored strain represents ~22% of the total free energy available from GTP hydrolysis (see *Figure 4—figure supplement 1d*), indicating that tubulin can convert chemical energy into mechanical work with an efficiency similar to other, more conventional molecular motors, such as kinesin, (*Howard, 1996*) myosin, (*Rief et al., 2000*) and dynein (*Gennerich et al., 2007*).

The curvature strain energy carried by curling protofilaments is easily sufficient to make a major contribution to the motility of isolated kinetochores and recombinant kinetochore subcomplexes in vitro (*Volkov et al., 2013*; *Akiyoshi et al., 2010*; *Asbury et al., 2006*; *Powers et al., 2009*; *Tien et al., 2010*). The mechanical work harnessed by these reconstituted couplers from a disassembling microtubule tip is the product of the opposing force, *F*, against which they move, multiplied by the distance moved, δ. The distance moved per

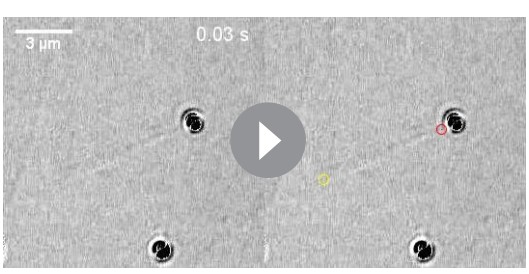

**Video 3.** Example of plucking force assay. A bead linked to the assembling plus end of a coverslip-anchored microtubule is subjected to increasing tension until the bead detaches. After bead detachment, the microtubule plus end disassembles, indicating that tubulin dimers were forcibly removed from the end. Two views of the same movie are shown. At right, the plus end and the coverslip-anchored portion of the microtubule are indicated by red and yellow markers, respectively.

released tubulin dimer, δ = 0.61 nm, is known from the structure of the microtubule lattice (i.e., the 8 nm length of an $\alpha\beta$-tubulin dimer divided by 13 protofilaments) (*Amos and Klug, 1974*). The maximum force against which a yeast kinetochore has been observed to track in vitro with disassembly is $F$ = 8.5 ± 1.9 pN (based on a line-fit to tracking speeds measured as a function of tension) (*Akiyoshi et al., 2010*). Multiplying these gives $F\cdot\delta$ = 5.2 ± 1.2 pN·nm of work per released dimer (equivalent to 1.3 ± 0.3 $k_BT$ per dimer, or 0.74 ± 0.17 kcal mole$^{-1}$), which is lower than our estimate of curvature strain in the protofilaments, by ~3 fold. Tip-couplers made by linking recombinant Dam1 complexes at high density to beads via long tethers can sometimes track against even higher forces, up to $F$ = 30 pN (*Volkov et al., 2013*). This maximum force corresponds to $F\cdot\delta$ = 18 pN·nm per released dimer (4.5 $k_BT$ per dimer, or 2.6 kcal mole$^{-1}$), a value nearly equal to our estimated protofilament strain.

The most directly comparable in vivo results are the classic microneedle-based measurements of Nicklas, who found that the total force required to stall anaphase chromosome movement in grasshopper spermatocytes was 700 pN (*Nicklas, 1983*, *1988*). Assuming this load was shared by 15 kinetochore-attached microtubules leads to an often-cited estimate of $F$ ≈ 50 pN per microtubule (*Nicklas, 1983*, *1988*). If all the energy driving chromosome movement was derived from disassembly of these kinetochore-attached microtubule tips, then the work harnessed per released dimer would be $F\cdot\delta$ = 30 pN·nm. Our results suggest that the majority of this energy could be derived from protofilament curvature strain.

For decades, the conformational wave model has remained a compelling but unproven hypothesis for kinetochore motility. By establishing that protofilaments carry enough curvature strain energy to make a major contribution to kinetochore movement, our findings lend strong support to the conformational wave hypothesis. Our study also reveals how the rate of microtubule disassembly can be altered dramatically by tubulin mutations that do not necessarily affect the amount of lattice strain or the intrinsic protofilament curvature. While the hyperstable mutant T238V tubulin that we examined here did not exhibit low lattice strain, we speculate that such low-strain mutants exist. We envision that the wave assay developed here will be useful for identifying low-strain tubulin mutants and for examining the mechano-chemistry of other recombinant tubulins.

## Materials and methods

### Expression and purification of yeast tubulin

Plasmids to express wild-type yeast $\alpha\beta$-tubulin with a His$_6$ tag fused to the C-terminus of $\beta$-tubulin were previously described (*Ayaz et al., 2014*, *2012*; *Johnson et al., 2011*). A plasmid to express the T238V mutation of Tub2p (yeast $\beta$-tubulin) was made by QuikChange mutagenesis (Stratagene), using an expression plasmid for wild-type Tub2 as template and with primers designed according to the manufacturer's instructions. The integrity of all expression constructs was confirmed by DNA sequencing. Wild-type or mutant yeast $\alpha\beta$-tubulin was purified from inducibly overexpressing strains of *S. cerevisiae* using nickel affinity and ion exchange chromatography (*Ayaz et al., 2014*, *2012*; *Johnson et al., 2011*) with the exception that the T238V mutant was eluted from the nickel-affinity column with 200 mM NaCl. T238V nickel elution fractions were treated with Universal Nuclease (Pierce) at room temperature for 1 hr prior to ion exchange chromatography. Tubulin samples for the laser trap assays were prepared at UT Southwestern, aliquoted and snap-frozen in storage buffer (10 mM PIPES pH 6.9, 1 mM MgCl$_2$, 1 mM EGTA) containing 50 µM GTP, shipped on dry ice to the University of Washington, and stored at −80°C.

### Wave assay setup

For each experiment, a small channel ~3 mm wide was formed by bonding a plasma-cleaned glass coverslip to a clean glass slide using two parallel strips of double-stick tape. GMPCPP-stabilized, biotinylated seeds were assembled from bovine brain tubulin, anchored to the coverslip surface, and then washed with a solution of 1 mM GTP in BRB80 (80 mM PIPES, 120 mM K$^+$, 1 mM MgCl$_2$ and 1 mM EGTA, pH 6.9) as previously described (*Akiyoshi et al., 2010*; *Franck et al., 2010*; *Powers et al., 2009*; *Franck et al., 2007*). Alternatively, for some experiments, axonemes purified from sea urchin sperm (*Waterman-Storer, 2001*) were adsorbed directly to the coverslips and then washed as above.

From the coverslip-anchored seeds or axonemes, dynamic microtubule extensions were grown and simultaneously decorated with anti-His-beads (prepared as described below) by introducing a suspension of the beads together with ~1 µM of either wild-type or mutant T238V yeast tubulin, freshly thawed, in microtubule growth buffer (BRB80 supplemented with 1 mM GTP, 5 mM DTT, 25 mM glucose, 200 µg mL$^{-1}$ glucose oxidase, 35 µg mL$^{-1}$ catalase) and then incubating the slide at 30°C for ~20 min. Only the microtubule minus ends were anchored to the glass coverslip – otherwise the filaments were unsupported.

To prepare the anti-His-beads, commercially available streptavidin-coated polystyrene microbeads (Spherotech Inc., Libertyville, IL) were functionalized by incubation of ~36 pM beads with 25 pM biotinylated anti-Penta-His antibodies (#34440, Qiagen, Valencia, CA) for 30 min, washed extensively, and stored at 4°C for up to several months. Just prior to each experiment, a small aliquot of the beads was incubated with a mixture of plain and biotinylated BSA (at 40 and 0.4 mg mL$^{-1}$, respectively) for >30 min, diluted into growth buffer with tubulin, and then used as described above. Pre-incubation with BSA was important for preventing non-specific attachment of the sparsely anti-His-decorated beads to the microtubules. Control experiments with beads lacking anti-His antibody confirmed that the attachments were specific after the BSA pre-incubation.

To ensure that most beads attached via single antibodies, the ratio of antibodies to beads during bead functionalization was kept very low, ~1:1, such that the fraction of beads under manual manipulation that would attach to the growing end of a microtubule was less than 50%. Active beads attached readily to growing ends, but not to the sides of microtubules. Their preference for growing ends was expected because the anti-His antibodies on each bead should become quickly and stably occupied by individual, unpolymerized (and His-tagged) tubulin dimers upon initial mixing of the beads and tubulin. Thus, bead-microtubule attachments presumably occurred via incorporation of bead-tethered tubulins into growing ends. Laterally attached beads, which are required for the wave assay, were nevertheless found readily after microtubule polymerization had commenced for ~20 min. These lateral attachments presumably arose by polymerization of microtubules past beads that were initially end-attached.

## Performing the wave assay

Our combination laser trap and laser scissors instrument has been described previously (*Franck et al., 2010*). Briefly, the microscope incorporates two lasers, a 1064 nm wavelength laser for trapping and a 473 nm laser for cutting microtubules. The trapping laser is focused to a diffraction-limited spot in the center of the field of view and the cutting laser is focused to an ellipse several micrometers away, to ensure that it does not interfere with trap operation. Both lasers are controlled independently by manual shutters. Individual microtubules can be severed by brief exposure to the cutting laser (<1 s).

To perform the wave assay, a suitable laterally attached bead was first selected and placed under laser trap tension. The bead-microtubule assembly was bent slightly upward, away from the coverslip, to prevent interactions between the bead and the coverslip, or between the disassembling tip and the coverslip, which would have interfered with the movements generated by the conformational wave. The distal, growing end of the microtubule was then severed using laser scissors to induce disassembly. The desired load was maintained by adjusting the position of the specimen stage under feedback control, implemented using custom software written in LabView (*Source code 1*). Significant forces could only be applied in the longitudinal direction, along the axis of the microtubule, because of the arrangement of the assay, with the beads tethered to flexible microtubule extensions anchored only by their minus ends to the coverslip. Flexibility of the unsupported microtubule extensions prevented the application of piconewton-scale forces in transverse directions. Bead-trap separation was sampled at 40 kHz while stage position was updated at 50 Hz to maintain the desired load for as long as the bead remained attached to the microtubule. The bead and stage position data were decimated to 200 Hz before storing to disk. Brief recordings (<20 s) were sufficient to capture the wave pulses. Up to 40 events could be recorded during a single 1 hr experiment, depending on the scarcity of suitably attached beads. Individual amplitudes and risetimes for all recorded pulses, as well as the means and standard errors for each measurement condition, are given in *Supplementary file 1*.

## Estimation of tether length

The beads in our experiments were tethered to the microtubules through a linkage that consisted of streptavidin on the bead surface, a biotinylated anti-Penta-His antibody (mouse monoclonal IgG1), and the C-terminal tail of $\beta$-tubulin, which is a disordered 30-amino-acid polypeptide segment exposed at the microtubule surface. Approximate lengths for streptavidin, 7 nm, and for the antibody, 18 nm, were estimated from PDB structures 1AVD and 1IGT, respectively. A length of 3.6 Å per amino acid was assumed for the C-terminal tail of $\beta$-tubulin, based on lengths measured for other mechanically unfolded polypeptides (e.g., see *Schwaiger et al., 2002*), yielding an estimate for the 30-amino-acid tail of 11 nm (excluding the $His_6$ tag, which is presumably bound up by the antibody). Adding these values for streptavidin, IgG, and the $\beta$-tubulin tail yielded a total of 36 nm for the complete tether. We considered deviations from this estimated length to examine how sensitively it would affect the predicted curves for stall force-vs-bead diameter and for unloaded amplitude-vs-bead diameter. The predicted curves were similar and fit our data well for tethers ranging from 30 to 42 nm (see *Figure 4*).

## Imaging of Bim1-GFP-decorated microtubules

Fluorescence imaging of growing microtubule tips decorated with Bim1-GFP was performed and analyzed as previously described, (*Geyer et al., 2015*) with the use of wild-type or T238V yeast microtubules grown in 1 mM GTP and in the presence of 50 nM Bim1-GFP.

## Methods for the plucking force assay

Beads for the plucking force assay were prepared and dynamic microtubule extensions grown from coverslip-anchored seeds exactly as described above for the wave assay. Single freely diffusing beads were selected and held near an individual growing microtubule end using the laser trap. Once the bead attached to the microtubule, increasing tension was applied with feedback control to maintain a constant loading rate, 0.25 pN s$^{-1}$, until the bead detached. Upon bead detachment, microtubule tip state was determined visually, from video-enhanced differential interference contrast (VE-DIC) images displayed live during the experiments. Usually, the microtubule began disassembling immediately after detachment of the bead (43 of 57 detachments, ~75%), which indicates that tubulin dimers were forcibly removed. Only detachments that were followed immediately by microtubule disassembly were included in the analyses of plucking force. The same stocks of antibody-decorated beads were used for the plucking force measurements with wild-type and T238V tubulin. Thus, the different plucking strengths cannot be attributed to different numbers of antibodies on the beads, or different types of antibodies, or different numbers of bonds between the beads and the microtubules. All the individual plucking force values are given in *Supplementary file 1* .

## Acknowledgements

This work was supported by a Sackler Scholars Award in Integrative Biophysics to JWD, by a Fellow Award from the Leukemia and Lymphoma Society to JWD, by an NIH Training grant to MEB (T32CA080416), by a Packard Fellowship to CLA (2006–30521), and by grants to CLA (NIH: RO1GM079373) and LMR (NIH: RO1GM098543 and NSF CAREER: MCB1054947). EAG was supported by an NSF Graduate Research Fellowship (2014177758).

## Additional information

### Funding

| Funder | Grant reference number | Author |
| --- | --- | --- |
| Sackler Scholars Program in Integrative Biophysics | | Jonathan W Driver |
| Leukemia and Lymphoma Society | | Jonathan W Driver |
| National Institutes of Health | T32CA080416 | Megan E Bailey |
| David and Lucile Packard | 2006-30521 | Charles L Asbury |

| Foundation | | |
| --- | --- | --- |
| National Science Foundation | Graduate Research Fellowship 2014177758 | Elisabeth A Geyer |
| National Institutes of Health | RO1GM098543 | Luke M Rice |
| National Science Foundation | Career Award MCB1054947 | Luke M Rice |
| National Institutes of Health | RO1GM079373 | Charles L Asbury |

The funders had no role in study design, data collection and interpretation, or the decision to submit the work for publication.

## Author contributions

JWD, Conceptualization, Investigation, Methodology, Writing—original draft; EAG, Conceptualization, Resources, Investigation, Methodology, Writing—review and editing; MEB, Investigation, Methodology, Writing—review and editing; LMR, Conceptualization, Resources, Supervision, Funding acquisition, Investigation, Methodology, Writing—review and editing; CLA, Conceptualization, Resources, Supervision, Funding acquisition, Investigation, Methodology, Writing—original draft, Writing—review and editing

## Author ORCIDs

Charles L Asbury, http://orcid.org/0000-0002-0143-5394

# Additional files

## Supplementary files

• Supplementary file 1. All individual pulse measurements. Individual amplitudes and risetimes for all recorded pulses, as well as the means and standard errors for each measurement condition, are given in the accompanying Excel spreadsheet. The spreadsheet also includes all the individual plucking force values.

• Source code 1. Custom software for controlling the laser trap.

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
