## [Decision Letter]

[Editors’ note: a previous version of this study was rejected after peer review, but the authors submitted for reconsideration. The first decision letter after peer review is shown below.]

Thank you for submitting your work entitled "Direct measurement of the force-generating capacity of protofilaments curling out from a disassembling microtubule tip" for consideration by *eLife*. Your article has been reviewed by three peer reviewers, and the evaluation has been overseen by a Guest Reviewing Editor (Marileen Dogterom) and a Senior Editor. The reviewers have opted to remain anonymous.

Our decision has been reached after consultation between the reviewers. Based on these discussions and the individual reviews below, we regret to inform you that your work will not be considered further for publication in *eLife*.

As you will see all three reviewers acknowledge that the work presented is technically sound, and that the results are carefully interpreted. However, the reviewers are not convinced that the new insights provided are substantial enough to warrant publication in *eLife*. They also have a number of additional (technical) questions that we hope will help you when preparing your work for publication elsewhere.

*Reviewer #1:*

The paper by Driver et al. describes a new optical tweezers-based assay to measure forces generated by outward curling protofilaments of depolymerising microtubules, as well as forces needed to detach both WT and mutant tubulin dimers. While the experiments are carefully performed and analysed, we have a number of concerns:

Our main concern regarding this manuscript is its novelty. The authors state in the Abstract that "whether curling protofilaments are strong enough to produce significant force has been unclear", however they later go on to cite literature on this subject that has already reported significant force from microtubule disassembly (Volkov et al., 2013; Grishchuk et al., 2005). In the main text the authors report that they have measured a force resulting from microtubule disassembly that "was at least 16‐fold higher than the maximum force measured previously(Grishchuk et al., 2005)". They fail to acknowledge however, that this 16-fold increase is due mainly to the calculations that the authors performed based on the geometry of the laser trap acting on the center of the bead bound laterally to a microtubule (see Figure 2 and Figure 2—figure supplement 2). These same geometrical considerations were previously applied to the force values obtained in Grishchuk et al., 2005, resulting in an increase of the estimated force on the bead surface that is well within the values reported in the manuscript by Driver et al. (see Grishchuk et al., 2005 Nature 438: 384‐388). Therefore, we believe that the bulk of the manuscript, although presenting a new experimental approach to measuring the bending force of tubulin protofilaments, does not report substantially novel results.

*Reviewer #2:*

This is a nice study using in vitro assays and optical trapping to measure the force generated by individual depolymerising microtubules. The authors use a neat design that pushes a microtubule attached bead 'sideways' when a microtubule depolymerises. The authors attribute this 'sideways' displacement to protofilament curls generating a force on the bead as they peel off the microtubule end. This is a plausible interpretation, since such curls have been often observed by electron microscopy. Using geometrical arguments they conclude that about a fifth of the chemical energy of GTP hydrolysis is transformed into mechanical work. The same work can be performed by a more slowly hydrolysable tubulin mutant, showing that kinetics can be uncoupled from the production of mechanical work. The authors also measure the force it takes to rip off tubulins from the microtubule end. Overall the experiments seem to be very carefully performed and the data and interpretations are convincing. Overall the results are very interesting and they agree well with several other published results that aimed at estimating the forces generated by microtubules at kinetochores.

Points of concerns/questions:

1) Some of the interpretations seem to depend critically on the assumptions that are made about the nature of the 'tether' between bead and microtubule, because it affects the lever effect. In the beginning of the Results the authors could be more explicit about what is the tether (His-tag on tubulin – biotinylated anti-His antibody – steptavidin bead). Why do they estimate that this tether is 36 nm long? What is the estimated error of this estimate of the tether length and how does this affect the precision of the conclusions?

2) The method describing the laser cutting does not seem to be described.

3) The authors state in the Methods that the antibody beads bind selectively to growing microtubule ends, which is maybe first a bit surprising, but they give a plausible explanation for that. However, no data are shown demonstrating that beads bind this way (apart from Figure 5 where the beads seem to be positioned close to MT ends using the trap). Since this bead-tubulin incorporation into the microtubule seems to be an important part of how the assay is set up, it might be worth mentioning this in the main text.

4) Reported forces are probably reported for the direction along the axis of the microtubule in the x/y plane. Correct? How well is that orientation known? Can the z-component of the force be measured?

5) The authors translate their measured h=20nm into observing a curled protofilament segment consisting of 4 tubulins. Why are the curls so short? Could it be that they are longer and less curved under load? Then this height might correspond to more tubulins.

*Reviewer #3:*

Driver et al. report on the direct measurement of the force-generating capacity of protofilaments curling out from a disassembling microtubule tip. While the 'conformational wave' model (according to which individual protofilaments pull on the kinetochore as they curl outward from a disassembling tip) is a popular explanation for motility powered by microtubule disassembly, the authors ask if curling protofilaments are indeed strong enough to produce significant force. To answer this question, they utilized a wave assay with recombinant yeast tubulin and a feedback-controlled laser trap. They show that curling protofilaments are surprisingly strong, enabling them to drive movement with forces and efficiency similar to conventional motor proteins. Taken together, the study demonstrates a direct way to examine tubulin mechanochemistry, thereby establishing that the wave mechanism can make a major contribution to kinetochore motility.

Experiments and data evaluation have been performed at the highest technical level. The described results are of excellent quality, leaving no doubt that the wave mechanism can generate enough force for chromosome separation. Overall, the manuscript is written in a most clear and concise manner.

I just have one major comment: Have the authors done a good enough job to differentiate their approach and their results from the seminal Grishchuk/McIntosh paper from 2005 (Grishchuk et al., 2005)? I do have the following questions:

a) What is the difference between the wave assays performed back then and now? Is the current study really demonstrating a "new" way to measure the forces associated with protofilament curling? Or, isn't the present work rather (just) an adaptation of the earlier approach?

b) How different are the current results from the earlier values? In Grishchuk et al., 2005 forces of 0.24 pN arising from curling of 1-2 protofilaments measured by 500 nm beads were reported. From these values, forces of about 5 pN on the bead (microtubule) surface – or 30 to 65 pN at the end of one depolymerizing microtubule – were estimated. These forces are quite substantial and I am not sure how fair the following statement is: However, the measured forces (~0.24 pN) were much weaker than forces generated at kinetochore-microtubule interfaces in vitro (9 pN or more) and also weaker than force estimates for kinetochore-microtubule attachments in vivo (4 to 8 pN during normal prometaphase) (Introduction, third paragraph). Doesn't it compare apples and pears? Consequently, I thought that also the following statements have already been answered by Grishchuk et al., 2005: But whether curling protofilaments are strong enough to produce significant force has been unclear (from the Abstract) and: The mechanisms underlying this type of energy transduction are unknown (from the Introduction). Perhaps I missed something but a clarification of the raised issues – and at places potentially toning down the novelty of the used approach and the drawn conclusions – seem appropriate.

[Editors’ note: what now follows is the decision letter after the authors submitted for further consideration.]

Thank you for resubmitting your work entitled "Direct measurement of conformational strain energy in protofilaments curling outward from disassembling microtubule tips" for further consideration at *eLife*. Your revised article has been favorably evaluated by Anna Akhmanova (Senior Editor and Reviewing Editor), and three reviewers.

The manuscript has been improved but there are some remaining issues that need to be addressed before acceptance, as outlined below:

1) It would be good if the authors could have another look at their text while keeping a broader audience in mind, for example when starting the Results section which is now maybe a bit technical. Some of this information, which mostly concerns the previous work, could probably be presented in a bit more conceptualised manner with the details being moved to the Methods or a supplementary Discussion.

2) The observation that the slowly-depolymerizing T238V mutant does not substantially change the amplitude-vs-force is surprising and interesting. This suggests that longitudinal interactions in the disassembling tip are not altered by the mutation. However, the authors don't have a way to directly confirm this. In the subsequent experiments, the authors acknowledge that the observed changes in Bim1 binding and "plucking" force of the mutant could be explained by changes in lateral or longitudinal contacts but cannot distinguish between these two possibilities with certainty. Can the authors discuss a bit more any potential structural differences between the wild-type and mutant, for example in protofilament number or the curvature of protofilament curls?

3) In the legend for Video 3, the word "coverslip" is misspelled. In the legend for Figure 3), the word "Initially" is misspelled.

---

## [Author Response]

[Editors’ note: the author responses to the first round of peer review follow.]

*As you will see all three reviewers acknowledge that the work presented is technically sound, and that the results are carefully interpreted. However, the reviewers are not convinced that the new insights provided are substantial enough to warrant publication in eLife. They also have a number of additional (technical) questions that we hope will help you when preparing your work for publication elsewhere.*

Summary – Why our work represents an important advance beyond previously published work:

1) Just *one* prior study has ever measured forces generated specifically by curling protofilaments. But the data were not sufficient to establish their fundamental force generating capacity, which depends on the strain energy they carry.

2) By overcoming technical limitations inherent to the prior study, we were able to measure protofilament strain energy directly for the first time. Doing so required us to modernize the wave assay, by engineering a site‐specific flexible tether between the laser‐trapped bead and the microtubule wall, and by applying force‐feedback.

3) Similar combinations of site‐specific protein engineering and sophisticated single molecule biophysical tools have revealed the mechanisms by which many vital protein machines function, including conventional motors, nucleic acid enzymes, and ion channels. Our study represents the first time that such a combination has ever been applied specifically to the study of protofilament curls.

4) By measuring the strain energy carried by curling protofilaments, our work establishes beyond doubt that they indeed have a large capacity to do mechanical work. It also enables comparisons with prior measurements of kinetochore motility to be made on the fundamental basis of energy, rather than force – a more rigorous approach that avoids assumptions about the structural underpinnings of kinetochore tip‐coupling, which remain poorly understood.

5) Using recombinant yeast tubulin enabled us to compare protofilament strain for wild‐type versus a hyperstable mutant for the first time. This comparison reveals, surprisingly, that disassembly speed can be uncoupled from curvature strain, a unique new insight into the fundamental mechano‐chemistry of tubulin.

*Reviewer #1:*

*The paper by Driver et al. describes a new optical tweezers-based assay to measure forces generated by outward curling protofilaments of depolymerising microtubules, as well as forces needed to detach both WT and mutant tubulin dimers. While the experiments are carefully performed and analysed, we have a number of concerns:*

*Our main concern regarding this manuscript is its novelty. The authors state in the Abstract that "whether curling protofilaments are strong enough to produce significant force has been unclear", however they later go on to cite literature on this subject that has already reported significant force from microtubule disassembly (Volkov et al., 2013; Grishchuk et al., 2005).*

We sincerely thank the reviewers for reading our manuscript and sharing their concerns. We firmly believe that our work, which is the first direct measurement of the conformational strain energy carried by curling protofilaments, represents an important advance over prior studies. However, we agree that our previous manuscript did not explain the novelty clearly enough. The new manuscript has been *extensively* revised and reorganized to make this clearer.

In particular, we explain more precisely how our work goes beyond the prior work of Grishchuk 2005 Nature (), which showed that disassembling microtubule tips can exert brief pulses of force against an attached bead. Their analysis suggested that curling protofilaments might be capable of generating very high forces, up to ~50 pN. However, the forces they actually measured were much lower (< 0.5 pN) and probably were restricted by interference of the attached beads with the short working strokes of the protofilaments. Displacement amplitudes were not reported. Because of these limitations, the energy carried by the curling protofilaments could not be determined. Our revised manuscript describes how we avoided these limitations to measure the wave energy. (See the sections entitled, “Modified Assay Improves Detection of Conformational Wave‐Driven Movement”, and, “Conformational Waves Carry Substantial Amounts of Strain Energy”.)

The other paper mentioned by the reviewer, Volkov 2013 PNAS, does not directly address forces or energies from curling protofilaments. Rather, it reports forces exerted by disassembling microtubules on couplers assembled from recombinant Dam1 complexes. Whether such couplers harness energy from curling protofilaments remains a matter of debate. An alternative, biased diffusion mechanism could also explain the results. Resolving this debate is indeed one of the reasons why it is important to measure the total energy carried by curling protofilaments. Moreover, protofilament strain energy is fundamental to microtubule dynamic instability. Thus, our measurements of protofilament strain provide insight into the fundamental mechano‐chemistry of tubulin.

*In the main text the authors report that they have measured a force resulting from microtubule disassembly that "was at least 16‐fold higher than the maximum force measured previously 24". They fail to acknowledge however, that this 16-fold increase is due mainly to the calculations that the authors performed based on the geometry of the laser trap acting on the center of the bead bound laterally to a microtubule (see Figure 2 and Figure 2—figure supplement 2).*

With respect, the reviewer has misinterpreted our intention here. There is a > 16‐fold difference between measured ‘raw’ forces – i.e., forces measured directly by the laser trap, and *not* adjusted by any calculations or geometrical considerations. Even with our largest beads (900 nm diameter) we observed pulses of motion against raw (unadjusted) forces up to 8 pN. With smaller beads, we saw pulses against even higher forces. Thus, the raw forces we measured were at least 16‐fold greater than the maximum (unadjusted) force measured in Grishchuk et al., 2005, which was only 0.46 pN. In our revised manuscript, we have made a clearer distinction between the forces actually measured in Grishchuk et al., 2005, and the much higher estimate they suggested based on their analyses. (See, for example, the third paragraph in our Introduction, and subsection “Conformational Waves Can Drive Movement Against Large Opposing Loads” in Results.) All the forces measured in our own experiments are reported as raw values, which was true in our previous submission but we have made it clearer in the new version.

We suspect that one of the reviewer’s primary concerns was a more general one – an overall concern about basing conclusions on large extrapolations. We fully share this concern. Indeed, it is one of the major limitations of the prior work (Grishchuk et al., 2005) that our work addresses in several ways: First, by engineering a flexible tether between the bead and the microtubule, we avoided interference of the bead with the curling protofilaments and avoided very high leverage, such that the trapping force was amplified only modestly at the bead surface (~2‐fold, depending on bead size; see subsection “Mechanism of Wave-Driven Movement: Protofilaments Push Laterally, Bead Pivots About Tether”, first paragraph).

Second, by using a feedback controlled laser trap we measured pulse amplitudes as functions of both trapping force and bead size to rigorously test the mechanism underlying bead movement in the assay (see the last paragraph of the aforementioned subsection).

Third, and most importantly, we measured the curvature strain energy in the protofilaments (subsection “Conformational Waves Carry Substantial Amounts of Strain Energy”). This strain energy is a more fundamental quantity than the magnitude of force produced, because it is what fundamentally limits the work output of the conformational wave mechanism. Grishchuk et al., 2005 does not report energies. Examination of their data shows that the energies they measured were certainly very small, but it is not possible to accurately estimate the energies from their experiments.

*These same geometrical considerations were previously applied to the force values obtained in Grishchuk et al., 2005, resulting in an increase of the estimated force on the bead surface that is well within the values reported in the manuscript by Driver et al. (see Grishchuk et al., 2005 Nature 438: 384‐388). Therefore, we believe that the bulk of the manuscript, although presenting a new experimental approach to measuring the bending force of tubulin protofilaments, does not report substantially novel results.*

We respectfully disagree. Our application of a force clamp and our use of flexible tethers between the bead and the microtubule allowed us to measure the full mechanical work output. Our directly measured work output (not adjusted by any calculations or geometric considerations) is much larger than measured in the previous study, and it is also independent of the size of the bead, confirming that we are indeed measuring the intrinsic capacity. Data from the previous study were insufficient to measure the energy of the wave, or to estimate the curvature strain energy per tubulin dimer.

To get past the dependence on assay geometry, it is important to make comparisons in terms of energy, rather than relying solely on comparisons of force. By estimating the energy per tubulin, our results can be compared not only with previous kinetochore tip‐coupling assays, but also with prior (and less direct) estimates of protofilament strain from thermodynamic, microtubule bending, and modeling studies.

*Reviewer #2: […] Points of concerns/questions:*

*1) Some of the interpretations seem to depend critically on the assumptions that are made about the nature of the 'tether' between bead and microtubule, because it affects the lever effect. In the beginning of the Results the authors could be more explicit about what is the tether (His-tag on tubulin – biotinylated anti-His antibody – steptavidin bead). Why do they estimate that this tether is 36 nm long? What is the estimated error of this estimate of the tether length and how does this affect the precision of the conclusions?*

All three reviewers wanted more clarity about this. The main text now describes our tethering scheme more explicitly (see subsection “Modified Assay Improves Detection of Conformational Wave-Driven Movement”, last paragraph) and the methods include a complete explanation of how our estimate of 36 nm was calculated (see subsection “Estimation of Tether Length”). Briefly, we estimate 11 nm for the β‐tubulin C‐terminal tail (30 amino acids at 0.36 nm each), plus 18 nm for the end‐to‐end length of the IgG (based on measurement in pymol of the PDB structure of an IgG), plus 7 nm for streptavidin (based on measurement in pymol of the PDB structure of avidin). We also note that our primary conclusions are not sensitive to the precise tether length, because the tether is long enough to avoid very high leverage. This point is now illustrated in Figure 4, which shows how the predicted curves for stall force‐vs‐bead size and for unloaded amplitude‐vs‐bead size do not change much when the tether length is adjusted by ± 6 nm.

*2) The method describing the laser cutting does not seem to be described.*

We thank the reviewer for their interest in the laser cutting method. We have added a brief summary of our use of a focused, 473‐nm laser to cut microtubules (subsection “Performing the Wave Assay”, first paragraph), which has also been described in a previous publication (Franck et al., 2010).

3) The authors state in the Methods that the antibody beads bind selectively to growing microtubule ends, which is maybe first a bit surprising, but they give a plausible explanation for that. However, no data are shown demonstrating that beads bind this way (apart from Figure 5 where the beads seem to be positioned close to MT ends using the trap). Since this bead-tubulin incorporation into the microtubule seems to be an important part of how the assay is set up, it might be worth mentioning this in the main text.

We felt that this detail about how the beads bind initially to growing tips and then, over time, the tip grows past them, was perhaps not so interesting except for those who wish to repeat our experiments, and thus we have explained it in the methods. We are certainly willing to move the explanation to the main text, if necessary. But we feel this would distract from the primary message of the paper.

*4) Reported forces are probably reported for the direction along the axis of the microtubule in the x/y plane. Correct? How well is that orientation known? Can the z-component of the force be measured?*

Yes, the reviewer is correct. The reported forces are always along the direction of the microtubule long axis, i.e. the x‐component. Microtubule extensions of at least several micrometers in length were required to perform the wave experiments, so that the bead‐microtubule assemblies could be bent slightly upward, away from the coverslip, to prevent unwanted interactions between the bead/microtubule and the coverslip. The extensions are very flexible in the transverse directions – they easily bend – and thus they cannot exert significant forces in either the y‐ or z‐directions. We have added an explanation of this point to the Methods (subsection “Performing the Wave Assay”, last paragraph). The orientations of the microtubules are easily determined during the experiment, both from VE‐DIC images displayed continuously at 30 Hz on a video monitor, and also from thermal motion of an attached bead, which our instrument tracks with high precision (nanometer scale) and displays continuously (at ~60 Hz) on the computer. The motion of a bead alone in the trap is isotropic in the x‐y plane. When tethered to a coverslip‐ anchored microtubule extension and placed under tension, the bead’s thermal motion is attenuated specifically along the microtubule long axis.

*5) The authors translate their measured h=20nm into observing a curled protofilament segment consisting of 4 tubulins. Why are the curls so short? Could it be that they are longer and less curved under load? Then this height might correspond to more tubulins.*

We have considered the possibility of larger curls, but this seems unlikely to us for two reasons. First, a curl height of h=20 nm is expected based on cryo‐EM images of protofilaments curling outward from pure microtubules in vitro. For example, McIntosh et al. (2008) report that protofilament curls in vitro have a mean contour length of 36 nm (for N = 56 curls), implying ~4.5 tubulin dimers per curl. The same paper reports 20° curvature per dimer, implying that the curl tips project from the surface of the microtubule, on average, by a height of h = 23 nm. The second reason larger curls seem unlikely to us is that our amplitude data are not well‐fit by models with substantially larger curls, as explained in the following paragraph.

We do agree that the protofilaments are probably being ‘straightened’ by the applied load. Indeed, we show that their curling motions are gradually suppressed as more opposing force is applied to them (Figure 2) and that they can be completely suppressed with enough opposing force (we refer to this maximum as the ‘stall force’; see Figure 4). Moreover, by extrapolating our amplitude‐vs‐force measurements to zero force, we estimate the unloaded amplitude when the curling is unopposed. As you can see below, curl heights much larger than 20 nm would not provide a very good fit to this data:

Author response image 1.Data recopied from Figure 4.Curves show predictions assuming a tether of 36 nm and a curl height, *h*, as indicated.**DOI:**
http://dx.doi.org/10.7554/eLife.28433.021

Curl heights of 18 to 22 nm fit the data well, implying curls with contour lengths ranging from 29 to 34 nm, which again suggests about four αβ‐tubulin dimers per curl.

*Reviewer #3:*

*[…] I just have one major comment: Have the authors done a good enough job to differentiate their approach and their results from the seminal Grishchuk/McIntosh paper from 2005 (Grishchuk et al., 2005)?*

Reviewers #1 and #3 both questioned the novelty of our work. We believe the revised version does a much better job explaining why our work is an important advance over the prior study.

*I do have the following questions:*

*a) What is the difference between the wave assays performed back then and now? Is the current study really demonstrating a "new" way to measure the forces associated with protofilament curling? Or, isn't the present work rather (just) an adaptation of the earlier approach?*

We firmly believe that our work, which is the first direct measurement of the conformational strain energy carried by curling protofilaments, represents an important advance over prior studies. However, we agree that our previous manuscript did not explain the novelty clearly enough. The new manuscript has been extensively revised and reorganized to make this clearer.

In particular, we explain more precisely how our work goes beyond the prior work of Grishchuk et al., 2005 Nature, which showed that disassembling microtubule tips can exert brief pulses of force against an attached bead. Their analysis suggested that curling protofilaments might be capable of generating very high forces, up to ~50 pN. However, the forces they actually measured were much lower (< 0.5 pN) and probably were restricted by interference of the attached beads with the short working strokes of the protofilaments.

Displacement amplitudes were not reported. Because of these limitations, the energy carried by the curling protofilaments could not be determined. Our revised manuscript describes how we avoided these limitations to measure the wave energy. First, by engineering a flexible tether between the bead and the microtubule we avoided interference of the bead with the curling protofilaments and avoided very high leverage, such that the trapping force was amplified only modestly at the bead surface (~2‐fold, depending on bead size; see subsection “Mechanism of Wave-Driven Movement: Protofilaments Push Laterally, Bead Pivots About Tether”, first paragraph). Second, by using a feedback controlled laser trap we measured pulse amplitudes as functions of both trapping force and bead size to rigorously test the mechanism underlying bead movement in the assay (see the last paragraph of the aforementioned subsection). Third, and most importantly, we measured the curvature strain energy in the protofilaments (subsection “Conformational Waves Carry Substantial Amounts of Strain Energy”). This strain energy is a more fundamental quantity than the magnitude of force produced because it is what fundamentally limits the work output of the conformational wave mechanism. Grishchuk et al., 2005 does not report energies. Examination of their data shows that the energies they measured were certainly very small, but it is not possible to accurately estimate the energies from their experiments.

*b) How different are the current results from the earlier values? In Grishchuk et al., 2005 forces of 0.24 pN arising from curling of 1-2 protofilaments measured by 500 nm beads were reported. From these values, forces of about 5 pN on the bead (microtubule) surface – or 30 to 65 pN at the end of one depolymerizing microtubule – were estimated. These forces are quite substantial and I am not sure how fair the following statement is: However, the measured forces (~0.24 pN) were much weaker than forces generated at kinetochore-microtubule interfaces in vitro (9 pN or more) and also weaker than force estimates for kinetochore-microtubule attachments in vivo (4 to 8 pN during normal prometaphase) (Introduction, third paragraph). Doesn't it compare apples and pears?*

With hindsight, we agree that comparison of conformational wave‐generated forces measured in Grishchuk et al., 2005 with those measured in tip‐coupling assays is confusing, since the geometries of the experiments are likely to be very different (“apples and pears”). The better comparison is between energies per released tubulin. In the revised manuscript, we compare energies rather than forces(Discussion, fourth and fifth paragraphs).

*Consequently, I thought that also the following statements have already been answered by Grishchuk et al., 2005: But whether curling protofilaments are strong enough to produce significant force has been unclear (from the Abstract) and: The mechanisms underlying this type of energy transduction are unknown (from the Introduction).*

We respectfully disagree. The 2005 Nature paper by Grishchuk (Grishchuk et al., 2005) leaves considerable doubt about whether curling protofilaments can produce enough force to make a significant contribution to the tip‐tracking motility of kinetochores. The Abstract of Grishchuk et al., 2005 makes a bold but, in our opinion, unsupported claim that, “a single depolymerizing MT can generate about ten times the force that is developed by a motor enzyme”. Later in the paper it is also speculated that 30 – 65 pN could be produced by a “well‐designed” coupling device.

However, the actual forces measured in this study were far lower, only 0.24 pN on average. Only ~0.46 pN is reported in the paper as “approximately the maximal force that this system can develop.” Thus, the statements that much higher forces can be generated were based on ~100‐fold extrapolations from the measured data. The truth of any statement based on such a large extrapolation can reasonably be considered uncertain, in our view.

While it is very clear that kinetochore complexes can harness considerable energy from disassembling microtubule tips, whether such couplers harness energy specifically from curling protofilaments remains a matter of debate. An alternative, biased diffusion mechanism could also explain the results. Resolving this debate is indeed one of the reasons why it is important to measure the total strain energy carried by curling protofilaments.

*Perhaps I missed something but a clarification of the raised issues – and at places potentially toning down the novelty of the used approach and the drawn conclusions – seem appropriate.*

The new manuscript has been extensively reorganized and rewritten. We believe it does a much better job of clarifying why it is important and what is novel with respect to the prior Grishchuk study.

[Editors' note: the author responses to the re-review follow.]

*The manuscript has been improved but there are some remaining issues that need to be addressed before acceptance, as outlined below:*

*1) It would be good if the authors could have another look at their text while keeping a broader audience in mind, for example when starting the Results section which is now maybe a bit technical. Some of this information, which mostly concerns the previous work, could probably be presented in a bit more conceptualised manner with the details being moved to the Methods or a supplementary Discussion.*

We agree that in some instances extra details were included in the Results, which were not absolutely required and perhaps distracting. In the revised draft, we have moved many of these technical details into the Materials and methods. We have also removed the first several sentences from the second section of the Results, which originally referred to the previous work, in order to better focus this section on our new findings. With these changes, we feel the manuscript is more streamlined and more accessible to a broad audience. Given the strong criticism of our first draft that it did not report substantially novel results, we feel it is important to be explicitly clear about how our work differs from, and goes beyond the previous work. Thus, we have chosen to retain some information about the limitations of the previous work and about the precise mechanism underlying bead movement in our work, in the first and third sections of the Results.

*2) The observation that the slowly-depolymerizing T238V mutant does not substantially change the amplitude-vs-force is surprising and interesting. This suggests that longitudinal interactions in the disassembling tip are not altered by the mutation. However, the authors don't have a way to directly confirm this. In the subsequent experiments, the authors acknowledge that the observed changes in Bim1 binding and "plucking" force of the mutant could be explained by changes in lateral or longitudinal contacts but cannot distinguish between these two possibilities with certainty. Can the authors discuss a bit more any potential structural differences between the wild-type and mutant, for example in protofilament number or the curvature of protofilament curls?*

We agree that the similar pulse amplitude‐vs‐force curves for T238V and wild‐type tubulin is surprising and interesting. We considered the possibility that the mutation might alter the curvature of protofilament curls, but our data suggest otherwise because altering protofilament curvature should affect the amplitudes (and energies) of the pulses. Pulse amplitudes should depend on *three* properties of the protofilament curls, their intrinsic curvature, their contour length, *and* their flexural rigidity. The similar amplitude‐vs‐force relation therefore suggests that *all three* properties are unchanged by the mutation. In the revised manuscript, we have edited both the Results (subsection “Disassembly Speed Can Be Uncoupled from Curvature Strain”) and Discussion sections (last paragraph) to make this point explicitly clear.

*3) In the legend for Video 3, the word "coverslip" is misspelled. In the legend for Figure 3), the word "Initially" is misspelled.*

Thank you for finding these typos. They are corrected in the new draft.